# Ultrathin Nano-Absorbers in Photovoltaics: Prospects and Innovative Applications

**Maximilian Götz \*** , **Norbert Osterthun, Kai Gehrke, Martin Vehse and Carsten Agert**

Urban and Residential Technologies, DLR Institute of Networked Energy Systems, 26129 Oldenburg, Germany;
norbert.osterthun@dlr.de (N.O.); kai.gehrke@dlr.de (K.G.); martin.vehse@dlr.de (M.V.);
carsten.agert@dlr.de (C.A.)
**\*** Correspondence: maximilian.goetz@dlr.de

**Abstract:** Approaching the first terawatt of installations, photovoltaics (PV) are about to become the major source of electric power until the mid-century. The technology has proven to be long lasting and very versatile and today PV modules can be found in numerous applications. This is a great success of the entire community, but taking future growth for granted might be dangerous. Scientists have recently started to call for accelerated innovation and cost reduction. Here, we show how ultrathin absorber layers, only a few nanometers in thickness, together with strong light confinement can be used to address new applications for photovoltaics. We review the basics of this new type of solar cell and point out the requirements to the absorber layer material by optical simulation. Furthermore, we discuss innovative applications, which make use of the unique optical properties of the nano absorber solar cell architecture, such as spectrally selective PV and switchable photovoltaic windows.

**Keywords:** ultrathin absorber; amorphous germanium; spectrally selective solar cell; switchable solar cell; absorption enhancement

## 1. Introduction

Thin film photovoltaics (PV) technology has been developed as an alternative to crystalline-silicon-wafer-based technology (c-Si), mainly because of concerns related to the return of energy and material restrictions [1]. Even though these restrictions are questionable today, it is still impressive that the ~100 μm thick c-Si absorber can be replaced by thin films of ~1 μm thickness [2]. This enormous reduction in thickness is possible, mainly because the direct bandgap semiconductor materials Cadmium Telluride (CdTe), Cupper Indium Gallium Selenide (CIGS), and amorphous Silicon (a-Si) possess substantially higher absorption coefficients compared to the indirect bandgap c-Si. The absorption coefficient of CIGS at a wavelength $\lambda = 1$ μm for instance is $\alpha_{CIGS}(1$ μm$) \sim 2 \times 10^4$ cm$^{-1}$ compared to that of c-Si being almost 100 times smaller $\alpha_{c\text{-}Si}(1$ μm$) > 3 \times 10^2$ cm$^{-1}$. This means that the desired reduction in the absorber layer thickness is compensated by an equally sized increase in the absorption coefficient.

Regardless of all complexity, a PV cell is built up from an absorber layer, electrical contacts with the ability to separate and extract photogenerated charge carriers, and light management structures. In the third generation of PV, different kinds of these light management structures are employed. Some of the structures are able to confine light within the absorber layer, meaning the devices are not restricted to single pass absorption anymore [3,4]. Recently, following the old mindset "thinner is better" and also out of scientific curiosity, the following question arose: how thin can the absorber layer be using light confinement structures? It turns out that, again, it can be made 100 times thinner compared to mainstream thin film technology by using ultra-high $\alpha$ absorber materials in combination with very strong light confinement in an optical cavity [5–7].

Absorption enhancement due to light confinement in optical cavities is already known from other photonic devices. It has been shown that the cavity effect can drastically increase the response and efficiency of photodetectors with graphene or carbon nanotube absorber layers [8,9]. Thin film structures called "super absorbers" show a point of perfect absorption due to the complete suppression of transmission and reflection [10]. Structural color filters and reflectors with much better thermal stability, a higher color gamut, and lower angle sensitivity, compared to dye-based filters, have been realized [11,12]. Additionally, several groups have demonstrated the potential of cavity-enhanced PV [5,6,13–15]. It quickly became clear that this new PV architecture brings about a new degree of freedom, the light confinement, with which the cell can be adapted for specific applications [16]. This is because in the cavity enhanced nano absorber PV, it is not the spectral distribution of the absorption coefficient that determines the optical properties of the PV cell, but it is the spectral response of the cavity. In the following, we will review the concept of the cavity enhanced nano absorber PV and show, by means of two applications, how the design of the optical cavity leads to innovative new PV solutions.

## 2. Methods and Materials

### 2.1. Experimental

The solar cells were deposited on commercial glass (10 cm × 10 cm) coated with an AZO front contact (thickness ~ 1 μm and mean square (RMS) surface roughness RMS ~ 10 nm). For optical characterizations, flat glass substrates were used. The substrates underwent cleaning with soap water and rinsing with deionized water and were dried with $N_2$ before the deposition. Silicon and germanium were deposited in one pump-down process in a capacitive-coupled plasma enhanced chemical vapor deposition PECVD capacitance reactor at 13.56 MHz frequency in a vertical multi-chamber inline system (Phoebus PECVD Lab C, Leybold, Alzenau, Germany). Hydrogenated intrinsic amorphous and microcrystalline silicon and germanium layers were deposited at 200 °C using $H_2$-diluted silane ($SiH_4$) and germane ($GeH_4$) precursor gases, respectively. Different n- and p-doping of the silicon electrodes was achieved by adding phosphine ($PH_3$) and diborane ($B_2H_6$), respectively. Detailed process parameters are summarized in Table 1. The substrates were heated up during the evacuation with additional 5 min temperature stabilization in the deposition chamber. A gas mixing phase of 1 min was applied before every deposition step. Hydrogen plasma passivation was performed after each layer deposition. Front contacts were exposed by laser ablation or by scratching of the absorber stack.

**Table 1.** Detailed PECVD process parameter for the ultrathin a-Ge:H solar cell. The deposition time for the germanium layer is $x$ = 500 or 1000 s for approximately 5 or 10 nm, respectively.

| Layer | Chamber | Pressure (mbar) | Power (W) | $SiH_4$ (sccm) | $GeH_4$ (sccm) | $H_2$ (sccm) | $B_2H_6$ (sccm) | $PH_3$ (sccm) | Time (s) |
|---|---|---|---|---|---|---|---|---|---|
| n a-Si | 3 | 4 | 70 | 20 | – | 450 | – | 18 | 48 |
| i a-Si | 3 | 4 | 70 | 34.7 | – | 300 | – | – | 26 |
| i a-Ge | 2 | 1.5 | 750 | | 6 | 286 | – | – | x |
| i μc-Si | 1 | 11 | 450 | 20.4 | – | 2000 | – | – | 14 |
| p μc-Si | 1 | 10 | 700 | 6 | – | 2000 | 4 | – | 42 |
| Hydrogen passivation | 1,2,3 | 4 | 70 | – | – | 600 | – | – | 60 |

The spectrally selective reflector was prepared by direct current (DC) magnetron sputtering at room temperature with 200 mm targets. A silver target was used as the metal (M) source and a ZnO:Al target with 0.5 wt.% $Al_2O_3$ doping (AZO) was used as the oxide (O) source. The films were deposited through a 1 cm × 1 cm mask without breaking vacuum in two chambers being part of a cluster tool type CS400PS (von Ardenne, Dresden, Germany). The process parameters are summarized up in

Table 2. The 400 nm AZO front contact for the spectrally selective cell was deposited at 220 °C in a Vistaris 600 (Singulus, Kahl am Main, Germany) by DC magnetron sputtering.

**Table 2.** Process parameters for MOMO fabrication.

| Process Parameter | Ag | AZO |
|---|---|---|
| Power [kW] | 0.2 | 1 |
| Pressure [mBar] | $8 \times 10^{-3}$ | $6 \times 10^{-3}$ |
| Distance target-substrate [mm] | 70 | 75 |
| Gas flow Ar/$O_2$ [sccm] | 60/– | 100/10 |
| Deposition rate [nm/s] | 1 | 1.45 |

Ag, Mg, Pd, and $MoO_x$ rear contact layers were deposited by electron beam evaporation in a box coater (DREVA LAB 450, VTD Vakuumtechnik Dresden GmbH, Dresden, Germany). Layer thickness and deposition rate were controlled in-situ by piezo electrical measurements. The parameters for the depositions are summarized in Table 3. All cells, except for the switchable one, were annealed after deposition at 130 °C for several hours.

**Table 3.** Deposition parameters of electron beam evaporation.

| Material | Voltage (kV) | Current (mA) | Rate (nm/s) |
|---|---|---|---|
| Ag | 8 | 35 | 1 |
| Mg | 6 | 2.9 | 0.5 |
| Pd | 8 | 49 | 0.05 |
| $MoO_x$ | 6 | 5.5 | 0.03 |

*2.2. Characterization*

Optical measurements were performed using a Cary 5000 spectrophotometer (Aglient Technologies, Santa Clara, CA, USA) with an integrating sphere. An Absolute Reflectance Transmittance Analyzer (ARTA Streulichtmessplatz, OMT Soultion BV, Eindhoven, Netherlands) enabled the angle dependent measurements. The angle of variation φ is defined by the surface normal of the sample. The measurement was done in steps of 15°. Reflectance and transmittance were measured for p- and s-polarized light, respectively. For evaluation, both were averaged. The *JV*-curves were measured with a WACOM dual lamp solar simulator (WXS-155 S-L2, WACOM, Tanaka, Japan) according to standard test conditions (AM1.5G spectrum, 1000 W/$m^2$, 25 °C). The switching process of Mg to $MgH_2$ was initiated by exposure for 15 min to 5% $H_2$ in $N_2$ at atmospheric pressure. Switching back to the initial state was reached by heating the sample for 15 min at 80 °C in dry air.

*2.3. Simulation*

The simulations in this paper were performed with the software package CODE/Scout (version 4.85). The 1D optical transfer matrix method was used to get reflection, transmission, and absorption values for the simulated layer stacks. Coherent light was assumed in the simulation of the thin films, while incoherent light was used for simulations of the thick glass substrate and the thick AZO layer. The optical models of the layers were developed for each single layer, respectively. Electrical field simulations were done with the same software package.

## 3. Nano Cavity PV

Ultrathin Resonant-Cavity-Enhanced PV terms the class of solar cells in which light absorption in an ultrathin semiconductor is enhanced by a resonant optical cavity. In the following, we review the recent work of our group in this field. First, we discuss the optical fundamentals of this cell concept and show which absorber materials are suitable for the use in nano cavities. Afterwards,

we present the recent results of novel applications, using ultrathin amorphous Germanium (a-Ge:H) as an absorber material.

### 3.1. Fundamentals of Cavity Enhanced Light Absorption

The absorption enhancement of thin film devices can be achieved by adjusting phase shifts in electromagnetic waves in such way that destructive interference is reached. The most famous example might be the asymmetric Fabry–Pérot cavity. A single dielectric layer with complex refractive index $\check{n}$ = $n_d + k_d$ and extinction coefficient $k_d$ close to zero, is deposited on a reflecting substrate. The layer stack has two interfaces that are important for our consideration: the interface one (I1) between the surrounding medium ($\check{n}$ = 1) and the dielectric material and the interface (I2) between the dielectric material and the substrate, as shown in Figure 1a. We are looking at an incident light beam from the surrounding medium on the dielectric. Partial waves, which penetrate through the dielectric and are reflected back from I2, experience a phase shift compared to the incident wave due to the optical path length in the medium. An additional phase shift of $\pi$ is added to the wave after reflection on the substrate. On the other hand, partial waves that are directly reflected at I1 only experience a phase shift of $\pi$ due to the interface reflection. A total optical path length difference (PLD) between the two partial waves can be calculated by PLD = $2d \times n_d \times \cos(\theta)$, where $d$ is the thickness of the dielectric and $\theta$ is the angle of incidence. This PLD is the phase shift of the partial wave compared to the wave reflected on the first interface. Constructive interference of light can be reached when the thickness of the dielectric fulfills the condition $d = m\lambda/4n_d$, where $m$ is an odd integer and $n_d$ the refractive index of the dielectric. The absorption enhancement in this type of resonant cavity depends completely on the optical path length inside the dielectric and is therefore restricted to rather thick layers. Assuming that the reflecting substrate shows no transmission, the absorption can be calculated by $A = 1 - R$, where the reflectance $R$ can be measured by experiments.

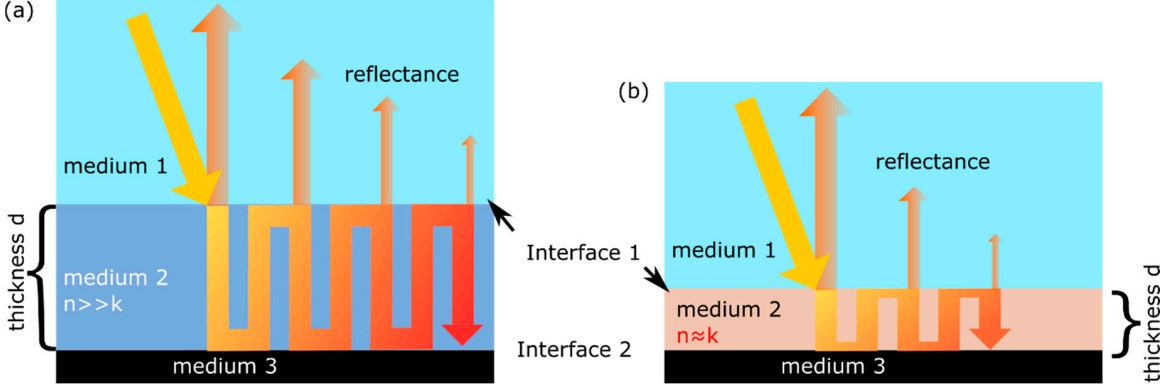

**Figure 1.** Absorption due to multiple internal reflections in a transparent dielectric (**a**) and an absorptive medium (**b**) on a reflective substrate.

Another condition for absorption enhancement can be found when the phase shift of the partial waves, which results from the reflections, becomes non-trivial, i.e., when it differs from 0 or $\pi$. This case is presented in Figure 1b. It can be reached by using an absorptive dielectric layer with $k_{Lossy} \sim n_{Lossy}$ [17,18]. In this case, interference conditions not only depend on the optical path length in the dielectric, but are increasingly dominated by the reflection phase shifts, especially for very thin absorber layers. Furthermore, in this configuration, broadband absorption is achieved, because the influence of the wavelength on the absorption is reduced. Destructive interference does not only depend on the optical path length, but also on the complex reflection coefficients of the interfaces. Absorption enhancement can be reached with thinner layers than $d = \lambda/4n_d$, if the phase shift of the reflection coefficients compensates the path length. This leads also to a reduced angle dependency of the cavity.

To explain the optical behavior of a nano-cavity, we consider the simplest system, a thin absorber layer on an optically thick reflective layer without transmission, as shown in Figure 1b. The absorption and reflection of this system depend strongly on the refractive index and extinction coefficient of the absorber layer medium. We have calculated the reflection of a 20 nm absorber layer with varying $n$ and $k$ values on a 300 nm silver layer in vacuum for wavelengths λ = 550 nm, λ = 600 nm and λ = 850 nm. Literature values and experimentally determined $n$, $k$- data of common semi-conductive materials are added to the plot to point out real materials in this simulation. The results are presented in Figure 2. It can be seen that the region of lowest reflection (perfect absorption) is restricted to a set of $n$, $k$ values. For larger wavelengths, this region shifts to lager n values, while the interval of the optimal extinction coefficient remains almost constant ($0 < k < 1.25$). For very high extinction coefficients ($k > 2.5$) only single pass absorption occurs and the cavity effect vanishes. Figure 2 clearly shows that not all materials are suitable for usage as ultrathin absorbers in this configuration. Materials like crystalline silicon (c-Si) have a low extinction coefficient, which means that most light is lost by reflection. Amorphous and micro-crystalline hydrogenated silicon (a-Si:H and μc-Si:H) have almost the same refractive index as c-Si, but show an increased extinction coefficient. This makes them suitable as nano-absorber material at λ = 550 nm and λ = 600 nm. At λ = 850 nm, the extinction coefficient of the Si-based materials is too small and the layer stack becomes too reflective. The same effect can be seen for GaAs-InAs layers. A truly broadband light absorption, like it is desired for photovoltaic applications, is achieved only with amorphous hydrogenated Germanium (a-Ge:H) and Molybdenum-Disulfide ($MoS_2$). Both of these materials show near unity absorption for λ = 550–850 nm. This makes them very interesting candidates to be used as ultrathin absorber layers for PV applications. It has to be noted that the refractive index of crystalline Germanium (c-Ge) is already too large for nano-absorber applications. In contrast to the c-Ge layer, the $CH_3NH_3PbI_3$ perovskite layer is also not suitable for this configuration because of its rather small n and k values.

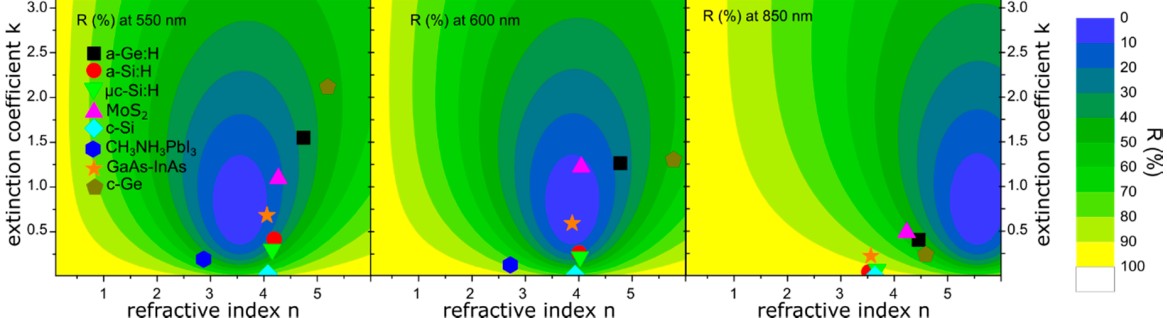

**Figure 2.** Influence of refractive index $n$ and extinction coefficient $k$ on the reflection of a 20 nm thick absorber layer on 300 nm Ag. The reflection has been calculated at three different wavelengths. The data points from literature for $MoS_2$ [19], c-Si [20], Perovskite ($CH_3NH_3PbI_3$) [21], GaAs-InAs [22], and c-Ge [23], as well as from fitted experimental data for a-Ge:H, a-Si:H, and μc-Si:H are included in the plot.

Notwithstanding this, this very simple configuration of a thin absorber on the reflective metal already shows the restrictions of the cavity effect, a full layer stack of a functioning solar cell is more complex. To study the influence of all the additional layers, which are necessary for the charge carrier separation and transport, we have calculated the optical behavior of three additional configurations (see Figure 3). The simple configuration on a glass substrate described above is shown in Figure 3a, while in Figure 3b p- and n-doped a-Si and μc-Si layers are added. The final layer stack, which is shown in Figure 3c, adds 80 nm ZnO:Al (aluminum-doped zinc oxide (AZO)) as a transparent electrode and represents a fully functional ultrathin solar cell. The respective calculated spectral reflection of the three configurations is shown in Figure 3d–f independent of the a-Ge:H layer thickness. The generated photocurrent density, which is presented in Figure 3g–i, is calculated by multiplying the absorption spectrum of the a-Ge:H layer for each layer thickness with the photon flux of the AM1.5 spectrum

and integrating over the spectral range λ = 350–1100 nm. It has to be noted that this is the upper limit of the current density, assuming lossless carrier extraction, and is just used as a tool to measure the absorption here.

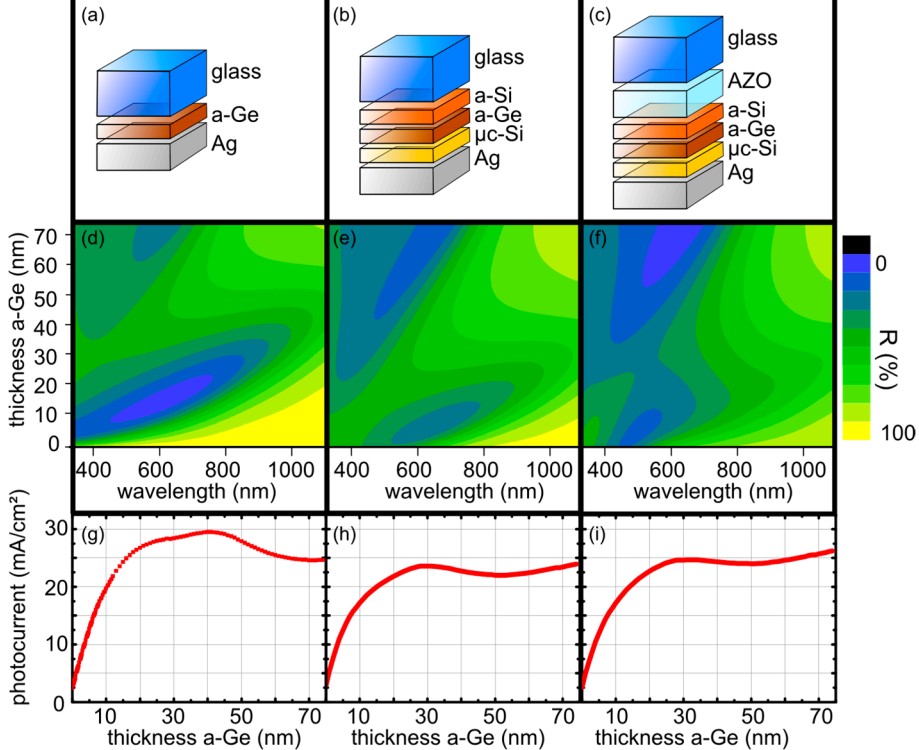

**Figure 3.** Influence of absorber thickness and surrounding materials on reflectance and absorption: layer stacks (**a**–**c**), reflection spectra (**d**–**f**), and photocurrent generated in the a-Ge:H absorber layer (**g**–**i**).

The reflection spectra of the simple layer stack (Figure 3d), the p–i–n stack (Figure 3e), and the complete device (Figure 3c) show comparable results. The first reflection minimum and absorption maximum is reached for Ge thicknesses between 1 and 20 nm. The absorption maximum shifts to larger wavelengths when the layer thickness is increased. Furthermore, the second order resonance absorption maximum can be seen for layer thicknesses above 30 nm. Since Si and Ge have a similar real part of the refractive index, the total optical path length of the absorber stack is increased when Si is introduced in the cavity. This makes the second absorption maximum more accessible with a thinner Ge absorber in the layer stacks of Figure 3b,c. Between the first and second absorption maximum, a region of broadband absorption can be found. It is striking to see, that the photocurrent generation density also mirrors the results of the reflection spectra. With increasing layer thickness, all three layer stack show first an increase in photocurrent. A peak value is reached for the simple layer stack (Figure 3g) at a thickness of 40 nm with almost 30 mA/cm$^2$. The more complicated devices reach their local maximum at a thickness of 30 nm with a photocurrent density of 24 and 25 mA/cm$^2$. The maximum values of photocurrent density can be found in the reflection plots in the region of broadband absorption between the two absorption maxima. A further increase in the thickness does not result in a significant enhancement of the photocurrent. This study of the layer thickness shows, that layers of a-Ge:H with less than a 10 nm thickness are able to reach high photocurrents and almost unity absorption in a realistic PV layer structure. Furthermore, it proves that the additional layers required for charge carrier separation and extraction do not significantly decrease the absorption in the device.

### 3.2. Amorphous Germanium Solar Cell

The simulations in the previous section showed the impact of the refractive index and layer thickness on the cavity effect. With this knowledge, the ultrathin resonant cavity solar cell has been fabricated. Figure 4a shows the layer stack. Between the n-doped a-Si:H (10 nm) and the a-Ge:H layer, a 5 nm thick intrinsic a-Si:H buffer layer is introduced. On the p-side of the cell, 5 nm of intrinsic μc-Si:H separate the a-Ge:H from the p-doped μc-Si:H (10 nm) layer. As front contact, a 1 μm AZO layer guarantees efficient charge carrier extraction. At the p-side of the cell, an Ag layer is used as rear contact. A detailed description of the fabrication process is given in the methods section of the publication. Since the defect density of a-Ge:H layers is rather high, carrier extraction decreases for thicker layers. Thus, we found the best a-Ge:H layer thickness to be around 10 nm. Figure 4b shows the JV-curves with the characteristic values of the solar cell. The open circuit voltage is $V_{oc} = 450$ mV and the short-circuit current density is $J_{sc} = 20$ mA/cm$^2$. This is slightly higher than that calculated for a layer thickness of 20 nm due to additional current generation in the intrinsic Si-buffer layers. The fill factor reaches FF = 58.7% which leads to an energy conversion efficiency of η = 5.32%. To the best of our knowledge, this is the highest value ever reported for any inorganic resonant cavity solar cell with a such thin semiconductor layer stack. In previously published studies our group demonstrated solar cells, which exploited the cavity effect in a-Ge:H and reached efficiencies of 2.08% [24] and 3.6% [6] in single junction devices and 4.0% [25] in a multi junction device. These results show that the p–i–n structure with a total thickness of only 40 nm can provide significant current density and that the cavity structure is a very promising way to realize efficient solar cells with ultrathin absorber layers.

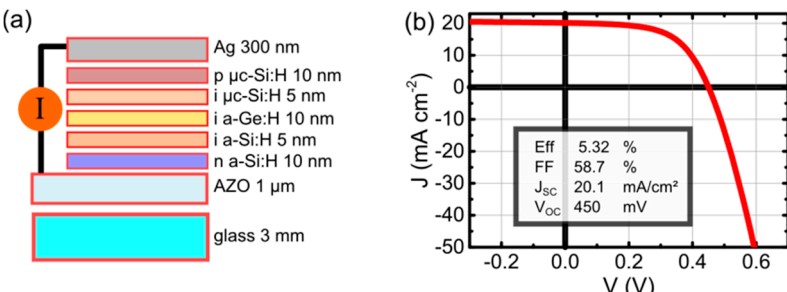

**Figure 4.** Layer stack (**a**) and *JV*-curve (**b**) of the reference solar cell with a 10 nm a-Ge:H absorber layer.

### 3.3. Applications for Photovoltaics with Ultrathin Absorbers

Due to the very unique optical properties of the resonant cavity enhanced solar cell, new applications far beyond standard PV are possible. Since the semiconductor layers are very thin, only a small fraction of light is absorbed in a single pass, making the device transparent in the first place. The absorption enhancement depends on the interface between absorber and rear contact. This opens the door for new applications, since the spectral behavior of the device can be designed by replacing the back reflector by functionalized layers. In the following chapters, two applications of the resonant cavity enhanced solar cell are presented, namely a switchable photovoltaic window and a spectrally selective solar cell. Figure 5a illustrates the concept of a switchable photovoltaic window. By using a back reflector that can be switched from a metallic reflective to a dielectric transparent state and back, the absorption of the device can be switched on and off. In the reflective state, the solar cell is opaque and works as the device presented in Figure 4, since the mirror enhances the local electric field inside the absorber. After switching the mirror to a transparent state, the absorption enhancement vanishes and light can pass the solar cell, rendering the window transparent. Figure 5b illustrates the concept of a spectrally selective solar cell as introduced in [26,27]. Here, a spectrally selective mirror, which is transparent in the blue and red wavelengths range, is used. The absorption enhancement is only present for the green and infrared light, where the mirror is reflective. This type of solar cell can be used for the combination of PV and photosynthesis.

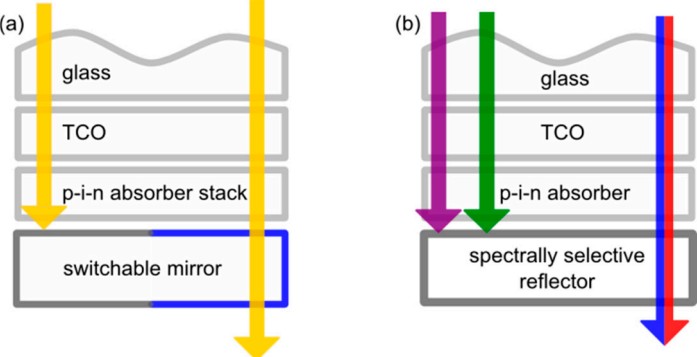

**Figure 5.** Illustration of a switchable solar cell (**a**) and a spectrally selective solar cell (**b**).

There is no doubt that other thin film technologies like CdTe or CIGS with absorber thicknesses above several 100 nm reach solar cell efficiencies far beyond the efficiencies presented in this publication. Nevertheless, the big advantage of the ultrathin solar cell is its variability for applications in semi-transparent PV. With a combined p–i–n layer stack thickness of less than 50 nm, the achieved efficiencies are a promising starting point to open up new areas for PV applications. The scope of this PV technology should rather be an addition to building elements like windows or façade than being used in PV power plants. Ultrathin solar cells might be a valuable addition to windows with low $\varepsilon$ coatings to improve their role from a passive to an active part of the building skin.

## 4. Switchable Photovoltaic Windows

Smart windows with the ability to turn opaque on demand have been available on the market for some years now. The devices use electrochromic coatings or liquid crystal films to block incoming light, when tinted. The blocked light is dissipated as heat to the outside of the building, remaining unused. Therefore, it would be favorable, if the light would be absorbed and used to generate solar power instead.

Switchable photovoltaics would allow for the conversion of the photon flux to an electrical current, while simultaneously blocking light from entering a building. The light absorption for photocurrent generation is reduced and light enters the building when the device is switched into a transparent state. Several approaches to realize such a switchable photovoltaic window already exist in literature [16,28–31]. They can be allocated to one of the following two categories: either the absorber material is switchable from an absorptive to a transparent state or the light management part of the layer stack is switchable, altering the absorption enhancement. In our case, a back reflector can be switched from a transparent state to a reflective state, increasing the absorption enhancement inside the optical nano cavity [6,16]. Several materials can be considered as a switchable mirror in contact with the ultrathin absorber. Liquid crystal switchable mirrors provide high reflectivity in one state and high transparency in the other state [32], but they require additional electrical contacts to initiate the switching process. Electrochromic mirrors [33,34] are another technology, which could be applied as switchable back contact. For the sake of simplicity, we decided to demonstrate the technology by using a gasochromic material, as shown in [35]. Thin magnesium films show a huge change of their optical properties by the absorption of hydrogen [36]. The material switches from a reflective metallic towards a dielectric transparent state. An advantage of this technology is the very high reflectivity and its applicability as an electrical contact for the solar cell in its metallic state. A palladium capping layer (5 nm) is applied on the Mg to protect it from water and oxygen. The Pd is known to serve as a catalyst, drastically improving $H_2$ uptake [37].

We simulated a switchable solar cell with Mg/Pd rear contact based on the resonant cavity enhanced technology shown above. Figure 6a,b shows a schematic drawing of the layer stack before (a) and after (b) hydrogen absorption. A layer stack of AZO (1 µm)/a-Si:H (10 nm)/a-Ge:H (5 nm)/µc-Si:H (10 nm)/Mg (25 nm)/Pd (5 nm) is used for the optical simulation. When Mg turns to $MgH_2$, the cavity

effect vanishes and the device becomes transparent. A thin a-Ge:H layer of only 5 nm was chosen, as this reduces the absorption of light in the transparent state of the cell, leading to higher transmission. A simulation of the electric field amplitude inside the device in both states is presented in Figure 6c,d. In the cavity "on" state, shown in Figure 6c, the electric field is confined inside the a-Si:H, μc-Si:H, and a-Ge:H layers over a broad spectral range. In particular, the a-Si:H layer is positioned at the highest field amplitude. In the a-Ge:H layer, the amplitude is slightly smaller. The electric field almost vanishes in the metallic Mg as it is decreasing exponentially. A small part of the field at wavelengths between 625 and 825 nm reaches the rear side of the Pd layer. This shows the remaining transmission of the field, which indicates that light is not completely confined in the absorber layer. It has to be noted that only the lower 25 nm of the AZO layer is shown in this field plot and that it has been set to be a non-coherent layer in the simulation to improve the readability of the graphs. After the hydrogen absorption of Mg to $MgH_2$, the electric field distribution changes drastically as can be seen in Figure 6d. Instead of being confined in the absorber layer, the field penetrates all layers, like it is expected for the cell in the transparent state. The cavity is switched "off". It has to be noted that parasitic absorption in $MgH_2$, Pd-H, as well as single pass absorption in the Ge and Si layers reduce the amount of transmitted light.

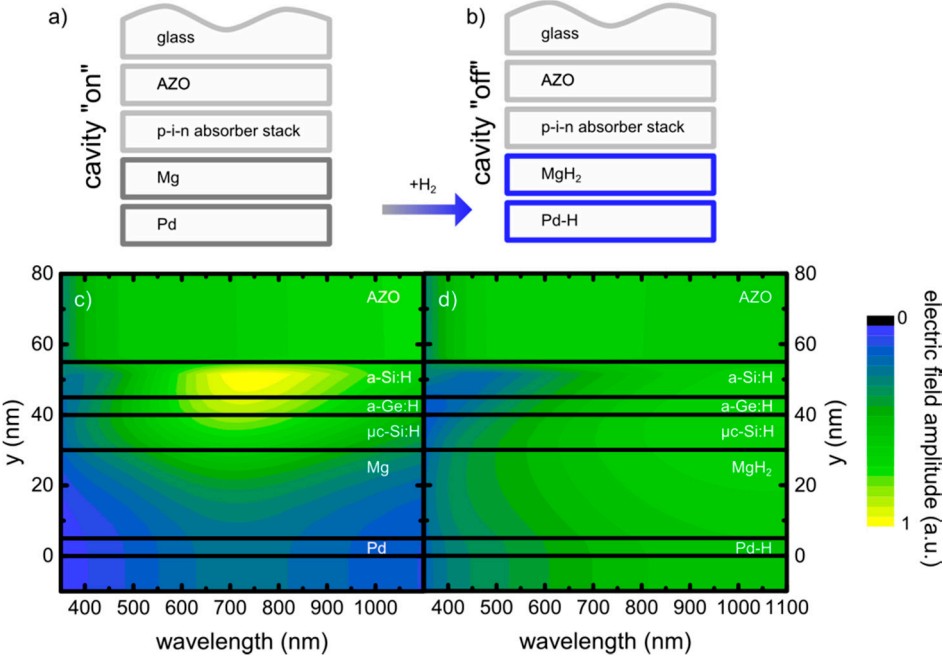

**Figure 6.** Switchable solar cell using Mg and Pd rear contact (**a**). Upon Hydrogen absorption, the mirror layers change to $MgH_2$/PdH (**b**). The simulated electric field amplitude inside the device in a cavity "on" (**c**) and a cavity "off" state (**d**).

The layer stack shown in Figure 6 has been realized experimentally to characterize the optical and electrical properties of the device, with two minor differences. The absorber layers are deposited with the same thicknesses as presented before. A 5 nm thick $MoO_x$ layer is added between the μc-Si:H and the Mg layer to improve the electrical contact between the p-doped μc-Si:H and Mg rear contact. $MoO_x$ is transparent due to its high bandgap and is used as a hole-selective contact [38]. The switching of the cell is initiated by exposure to 5% hydrogen in nitrogen gas at atmospheric pressure and room temperature for 15 min.

Figure 7a presents the measured transmission (*T*) of a realized solar cell stack in cavity a "on" and "off" state, as well as a sample without any rear contact. In the cavity "on" state, the transmission stays below a value of $T = 10\%$. After the hydrogen absorption of the Mg and Pd layer, the transmission increases up to a peak value of $T \approx 30\%$ at a wavelength of 746 nm. In the visible spectral range, the transmission rises to $T = 16\%$ at 500 nm and to $T = 27\%$ at 630 nm. As a reference, we measured

a cell without any rear contact, which reaches a transmission of $T = 56\%$ at a 840 nm wavelength. The reduced transmittance of the stack with hydrogenated magnesium back contact, compared to that without back contact, is most probably caused by parasitic absorption in the $MgH_2$ and Pd-H layers, as well as additional reflection losses. The interference peaks in the visible spectrum can be attributed to resonances in the AZO front contact. The hydrogen desorption process is initiated by exposure to ambient air. After several minutes, the solar cell returns to its absorptive state. Hydrogen desorbs from the thin switchable layer and reacts with oxygen from the surrounding air. This process can be accelerated by heating the device.

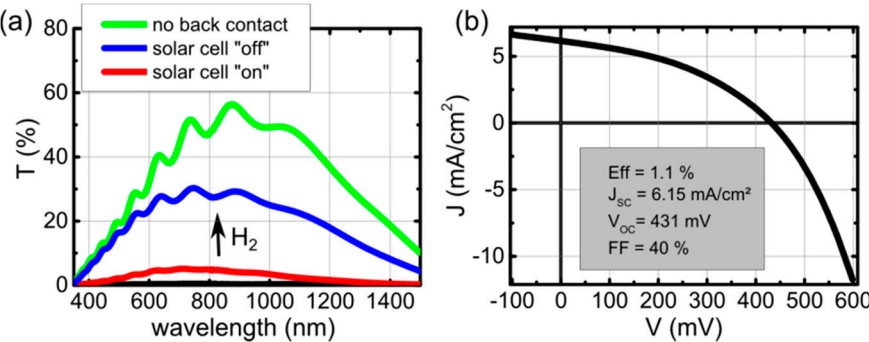

**Figure 7.** Realization of switchable solar cell: transmission measurements of cells in a cavity "on" and cavity "off" state (**a**). As a reference, a cell w/o back contact is shown. The *JV*-curve of the cell in cavity "on" state (**b**).

The solar cell characterization of the electrical parameters has been carried out using AM1.5 illumination in a solar simulator. The resulting *IV*-curve of a cell with switchable Mg/Pd rear contact is shown in Figure 7b. The cell in the opaque state reaches a short-circuit current density of $J_{SC} = 6.15$ mA/cm$^2$ and an open-circuit voltage of $V_{oc} = 431$ mV. This results in an efficiency of 1.1% with a fill factor (FF) of 40%. The efficiency of the cell is difficult to characterize after hydrogen is absorbed in the back contact due to the fact that only a limited amount of the generated charge carriers can be extracted by the non-conductive $MgH_2$ layer. The efficiency may be further increased by a better band alignment of the Mg rear contact with the doped layers as well as a further optimization of the optical field in the absorber layer.

The presented results show that the ultrathin cavity solar cell is the perfect base to develop a switchable photovoltaic window. By simply replacing the Ag back contact with a Mg/Pd (25 nm/5 nm) layer stack we were able to realize a functional solar cell with switchable transparency and light absorption. The cavity is switched "on" and "off" by the modification of the back contact from a reflective or transparent state, respectively. Further improvements in electrical and optical properties will lead to applications of this technology in larger scale devices such as switchable photovoltaic windows. For applications in real-life scenarios, the overall transparency should be increased and the efficiency further improved. To avoid hydrogen gas in the building facade, the gasochromic mirror could be replaced by a more sophisticated electrochromic mirror. This would allow the switching processes to be controlled with an electric potential instead of using gases. The possibility to dynamically change the transparency of a window and at the same time generate electricity could be a valuable addition to large window fronts.

## 5. Spectrally Selective Solar Cells

The combined use of solar radiation for photovoltaic and photosynthetic energy conversion, termed agrivoltaics, was suggested by Goetzberger and Zastrow already in 1982 [39]. Recently, this field of research gained a lot of interest, and several concepts have been demonstrated and tested under environmental conditions [40,41]. The majority of these projects use evenly spaced lines of PV modules installed above agricultural areas. The spacing between the modules leads to moving shadow

positions throughout an entire day. This allows all plants below the solar panels to get enough light to grow and thrive. Especially in semi-arid and arid regions, the highest potential of this technology is expected since the intense solar radiation, as well as the excessive water evaporation associated with it, might be reduced [42,43]. In contrast to the installation of opaque modules, spectrally selective solar cells (SSSC) can be used as an alternative approach, where no spacing is necessary [44–46]. In this concept, the spectral splitting of the solar radiation into a red and blue part of light for photosynthetic biomass production and a green and infrared part for the photovoltaic power generation is used. Within this, red and blue light is transmitted through the cells, while the remaining light is absorbed or reflected. Such a spectral selection makes sense, since the chlorophyll molecules, which are driving the photosynthesis, only absorb in the blue and red part of the spectrum, leaving the rest of the light unused (see Figure 8) [47,48]. Furthermore, it has been suggested that the spectral splitting can lead to improved thermal stabilization [45]. The full area applicability of SSSC modules makes the technology advantageous for the application in greenhouses, vertical gardening installation, and even photo bio reactors.

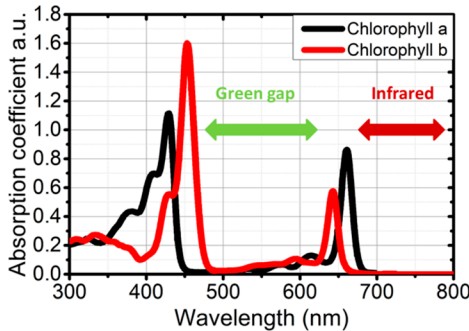

**Figure 8.** Absorption coefficient of chlorophyll a and b with the green gap and the infrared gap, modified from [49].

Spectrally selective PV modules have already been realized using organic solar cells and luminescent solar concentrator (LSC) technology [44,46]. By applying the LSC technology to greenhouses, significant growth in tomatoes and algae has been reported [44,45,50]. This demonstrates that plant cultivation is possible with red and blue light illumination. With the technology of the ultrathin resonant-cavity-enhanced solar cell, SSSC can be realized using only well-established, industry-proven thin film deposition processes. Moreover, the spectral selectivity can easily be tuned by only adjusting layer thicknesses [25]. In order to design an SSSC, we replace the opaque Ag back reflector of the ultrathin a-Ge:H reference cell by a spectrally selective mirror, as shown in Figure 9. Since the reflector represents one of the electrical contacts of the solar cell, it has to be conductive and must be applicable for charge carrier extraction. This can be realized with a multilayer reflector consisting of ultrathin semitransparent metal layers and transparent conducting oxide (TCO) layers. The materials that we use in our Metal–Oxide–Metal–Oxide layer stack (MOMO) are silver (Ag) and aluminum-doped zinc oxide (AZO). The MOMO reflector shows Fabry-Perot resonances due to the superposition of partial light waves inside the inner oxide layer sandwiched between the two metallic Ag layers. The peak positions of the resonances are determined by the optical thickness of this AZO layer [25,51]. A fine adjustment of the peak positions can be achieved by varying the thickness of the outer AZO layer that additionally serves as a protection for the outer Ag layer. The MOMO can be optimized to transmit blue and red light and reflect the green and infrared part of the spectrum as shown in Figure 9. The reflector shows transmission maxima in the red and blue spectral region with $T_{blue} \sim 60\%$ and $T_{red} \sim 75\%$, matching the chlorophyll absorption peaks. Green and near infrared light is reflected with up to $R_{green} \sim 80\%$, which demonstrates the huge potential of the MOMO layer stack for use in an SSSC. The inset in Figure 9 demonstrates the transmission and reflection of such a

reflector under white light illumination. In reflection the green color and in transmission the expected purple color is clearly visible.

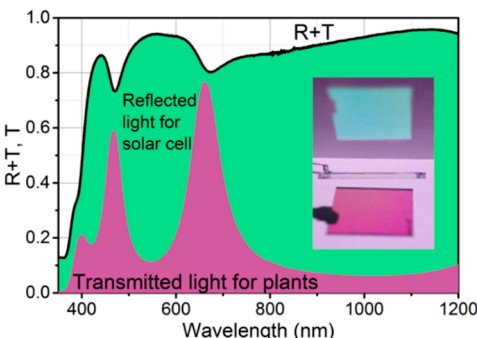

**Figure 9.** Reflection R and transmission T of the spectrally-selective reflector. Photons from the green area can be converted in the solar cell, while the violet area is transmitted to the plants. The white area marks the parasitic absorption. The thickness of the silver layers is 15 nm, while the sandwiched aluminum-doped zinc oxide (AZO) thickness is 271 nm and the thickness of the outer AZO is 85 nm. The inset shows a photograph of the Metal–Oxide–Metal–Oxide (MOMO) reflector on glass under white light illumination with its spectrally selective reflection and transmission colors.

This MOMO reflector was used to replace the opaque silver mirror of the ultrathin resonant-cavity-enhanced solar cell, which leads to a drastically altered absorption and transmission spectra of the cell. Absorption enhancement due to the cavity effect is only present for spectral regions where the MOMO layer stack is reflective, while in the remaining regions (red and blue) only single pass absorption occurs. We use a p–i–n solar cell stack based on amorphous silicon and germanium with an approximately 5 nm thick a-Ge:H absorber as described above. To avoid interference fringes related to the AZO front contact, we decreased its thickness to $d_{\text{frontAZO}} \sim 400$ nm. Figure 10a shows the illuminated IV curves of a solar cell with spectrally selective back contact compared to a cell with an opaque silver back contact. The extracted photovoltaic parameters are listed in the inset. Using the ~400 nm front contact and ~5 nm absorber layer, the opaque reference cell shows an efficiency of $\eta_{\text{ref}} = 3.6\%$, while the SSSC reaches an efficiency of $\eta_{\text{SSSC}} = 1.9\%$. For both cells, the open circuit voltage reaches $V_{\text{oc}} = 546$ mV. It can be concluded that the silver mirror and the MOMO reflector have comparable efficiencies regarding the charge carrier extraction. The short-circuit current density reaches $J_{\text{sc}} = 6.9$ mA/cm$^2$ for the SSSC, leading to a fill factor of FF$_{\text{SSSC}} = 50.1\%$, compared to that of the opaque cell, reaching $J_{\text{sc}} = 10.98$ mA/cm$^2$, leading to a FF$_{\text{ref}} = 60.8\%$. The reduced fill factor can be explained by the higher sheet resistance of the MOMO electrode, compared to the thick Ag layer. The current density of the SSSC is reduced as expected since less light is confined in the cavity. The optical transmission spectrum of the SSSC is shown in Figure 10b. As expected from the results of the MOMO reflector on glass, the two transmission peaks in the blue and red spectral range can be found. The transmittance of red light reaches a value of almost 55%, while the transmittance of blue light is slightly above 10%. The smaller transmittance in the blue spectral range is most probably the result of parasitic absorption within the doped silicon layers, which are used for charge carrier separation. Applying alternative contact schemes with wide bandgap materials could improve the optical performance in this case [52]. Moreover, as can be seen in Figure 10, 40% of the green and near-infrared light is still reflected instead of absorbed. Here, further improvement has to be achieved by reducing reflection losses. Within this, only the reflection of infrared light could be beneficial for enhanced thermal stabilization and might be interesting for regions, where greenhouses have to be cooled [45].

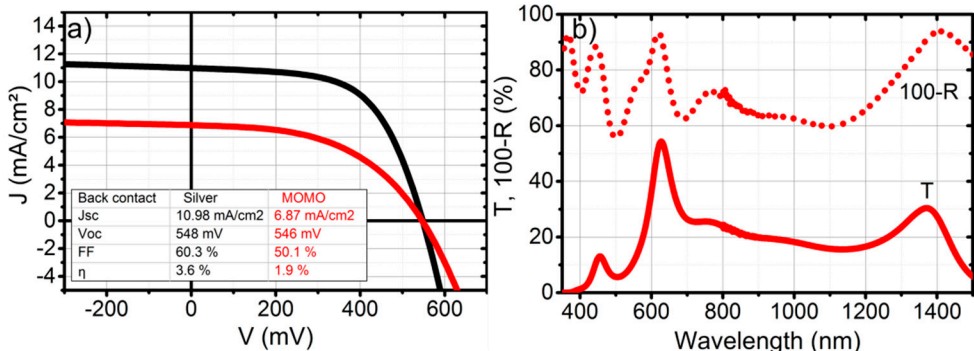

**Figure 10.** Comparison of *IV*-curves (**a**) for an opaque (black) and a spectrally selective solar cell (red) with a-Ge:H thickness ~5 nm. A corresponding transmission and reflection spectrum of the spectrally selective solar cell (**b**).

For the integration of SSSC in greenhouse roofs or facades, it is important that the angular dependence of the spectral selectivity is small. The spectral position of the transmission peaks and the transmitted intensity should ideally stay constant for all angles of incident light. In Figure 11, the angular performance of our SSSC is shown. The peak position of the transmitted blue light does hardly shift with an increasing angle of incident φ, although the peak intensity decreases from 15% to 5%. The position of the red peak shows a more pronounced shift towards smaller wavelengths for increased φ. Due to the angle stability, the transmission at λ = 650 nm remains at T ~ 40% for an illumination angle of φ = 50°. The infrared peak shows the highest shift towards smaller wavelengths, but this does not affect the absorption in chlorophyll. In addition, the reflection of the SSSC confirms the results of the transmission measurement. This is in good agreement with previously shown angle stabilities for Fabry–Perot resonators [16].

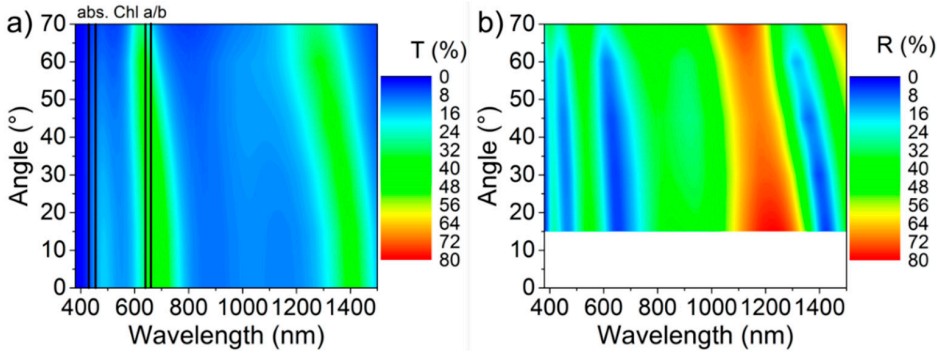

**Figure 11.** Color plots of the angle dependence of a spectrally-selective solar cell. Transmission T with the absorption maxima of chlorophyll a/b (**a**) and the corresponding reflection R (**b**).

Further improvement could be achieved by optimizing the back reflector for an optimal transmission spectrum at a higher illumination angle. This could be realized by adjusting the peak positions and shapes by changing the thickness of the different AZO and Ag layers. Another approach is the tuning of the illuminating conditions or to apply anti reflection coatings [53,54]. By using structured surfaces, the angle distribution of the incoming light can be optimized with regard to the angular interval of ±50°.

In conclusion, replacing the opaque back mirror of the resonant-cavity-enhanced solar cell by a Fabry-Perot-type spectrally selective mirror offers the possibility to realize a spectrally selective solar cell. While blue and red light is transmitted and can be used for plant or algae cultivation, the remaining light of the solar spectrum is used for photovoltaic electricity generation with no drawback concerning the open circuit voltage. The technology is very promising, since it is based on the well-established deposition techniques (plasma enhanced chemical vapor deposition (PE-CVD)

and magnetron sputtering), which are known to be scalable to large areas. Furthermore, the ability to tune optical resonances of the MOMO reflector makes it very unique, and the transmission peaks can be adjusted to other light harvesting molecules besides chlorophyll.

## 6. Conclusions

In this paper we reviewed our status on ultrathin resonant-cavity-enhanced solar cells and its novel applications in semi-transparent PV. We have shown the fundamental optical restrictions of the cavity effect and which common semiconductor materials can be used to reach broadband absorption enhancement. The simulation of layer stacks, using a-Ge:H as the absorber material, showed the enormous potential for photocurrent generation and that the absorption enhancement can be altered by changing the light confinement. The experimental realization of the solar cells confirmed that considerable efficiencies are reached and that switchable photovoltaic windows as well as spectrally selective solar cells can be realized with this technique.

Due to the wide range of applications of ultrathin absorbers in a resonant cavity shown here, we conclude that this publication can be the start for further intense research into this topic. Overall, this study strengthens the idea that ultrathin absorbers in a resonant cavity are a key technology for new PV applications for buildings and in combination with agricultural use.

**Author Contributions:** Conceptualization, M.G., N.O., K.G. and M.V.; methodology, N.O. and M.G.; software, M.G. and N.O.; validation, M.G., N.O., K.G. and M.V.; formal analysis, M.G. and N.O.; investigation, M.G. and N.O.; resources, M.V. and C.A.; data curation, M.G., N.O.; writing—original draft preparation, M.G., N.O. and K.G.; writing—review and editing, M.G., N.O., K.G. and M.V.; visualization, M.G. and N.O.; supervision, K.G., M.V. and C.A.; project administration, M.V. and C.A.; funding acquisition, M.V. and C.A. All authors have read and agreed to the published version of the manuscript.

**Funding:** This research received no external funding.

**Acknowledgments:** The authors thank D. Berends for sputter deposition of AZO layers and for helpful discussions. The authors also thank N. Neugebohrn, C. Lattyak and H. Meddeb for helpful discussions.

**Conflicts of Interest:** The authors declare no conflict of interest.

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
