# Peer review of "Ultrathin Nano-Absorbers in Photovoltaics: Prospects and Innovative Applications"

_coatings, doi:10.3390/coatings10030218_

Round 1

Reviewer 1 Report

In this article the authors review the state of the art on ultra-thin resonant-cavity-enhanced solar cells and their new applications in semi-transparent photovoltaic cells. The article is interesting and very well written. I recommend it to be accepted after the authors have addressed the minor comments indicated below.

1. On section 3, "Switchable Photovoltaic Windows", the authors demonstrate the technology by using a gasochromic material. They explain clearly the transition "opaque -> transparent" by saying that when Mg turns to MgH2, the device becomes transparent and the cavity effect vanishes. However, it is not clear how the cell will switch from transparent (MgH2) to opaque Mg. That is, it is not clear how in practice the dehydrogenation of MgH2 (transparent) -> Mg (opaque) can be achieved. Although this gasochromic based technology might be very interesting from an academic perspective, it is not clear how it can become feasible and interesting from a commercial perspective. It would be nice if the authors could develop a bit on these topics by adding a few sentences.

2. On the beginning of section 2.1 "Fundamentals of cavity enhanced light absorption" the authors explain some fundamental concepts and refer for example to the interface one (I1) and interface (I2) between dielectric material and substrate. In my opinion the authors should include an additional Figure (Figure 1 in this case) to explain these basic concepts more clearly. Although this topic is very relevant to the general OPV community, a significant part of this community is not very familiar with it. In this sense, an introductory figure explaining basic concepts of optical cavities would make the article more accessible and readable to non-experts in optical cavities (and consequently more citable). A Figure such as Figure 1c in Organic Electronics 15 (2014) 1545 and Figure 1e in Applied Energy 235 (2019) 1505 are just two possible examples.

3. Some misspellings: exemplarily (line 117); JSC and VOC (line 280); dfrontAZO (line 359)

Reviewer 2 Report

Please, find the attached file.

Author Response

We would like to thank the reviewer for taking the time to read our manuscript and give insightful comments on our presented work. We appreciated all comments and addressed them in the resubmitted version of the manuscript.

Please find below a point-by-point response to all of your comments, as well as the changes we made to the manuscript.

Comment #1: “I am concerned about the applicability of these nano-absorbers in the real world since as the
authors mentioned the efficiency of these devices is low compared to other thin film solar cells, for example CdTe or CIGS solar cells. What would make these device structures compelling in the real world? It would be good to add some discussion about it.”

We thank reviewer 1 for this insightful comment. To address the applicability of the ultra-thin absorbers in real live scenarios, we added an additional paragraph to our manuscript.

Changes to the manuscript /Additional text

(line 216)

There is no doubt that other thin film technologies like CdTe or CIGS with absorber thicknesses above several 100 nm reach solar cell efficiencies far beyond the efficiencies presented in this publication. Nevertheless, the big advantage of the ultra-thin solar cell is its variability for applications in semi-transparent PV. With a combined p-i-n layer stack thickness of less than 50 nm, the achieved efficiencies are a promising starting point to open up new areas for PV applications. The scope of this PV technology should rather be an addition to building elements like windows or façade than being used in PV power plants. Ultra-thin solar cells might be a valuable addition to windows with low ε coatings to improve their role from a passive to an active part of the building skin.

Minor Corrections: “I have very few minor corrections to the manuscript. Some of axis labels need to be improved to make it clear for the readers. For examples, Figure 1. It is good to correct the subscripts in the manuscript. For example, MgH2 throughout the manuscript, current density JSC, open-circuit voltage VOC etc in page 8.”

We thank the reviewer for raising these points and especially for finding the mistakes in the subscripts. We need to apologize for this. We improved the axis of Figure 1and corrected all misspellings and and wrong subscripts throughout the manuscript.

Reviewer 3 Report

The manuscript "Ultra-thin Nano-absorbers in Photovoltaics: Prospects and Innovative Applications" by Gotz et. al. describes some approaches to address new photovoltaics applications. The results are interesting, and this field continues to represent a significant direction for work in photovoltaics. My recommendation is that this manuscript could be suitable for publication in Coatings.

Author Response

We would like to thank the reviewer for taking the time to read through our manuscript and for the encouraging comment.

Changes to the manuscript:

We corrected all misspellings and subscripted all letters where necessary (for example: JSC->JSC). Furthermore, we tried to improve English sentences for improved readability.